# Disparate energy sources for slow and fast Dansgaard-Oeschger cycles

Diederik Liebrand[1, 2, 3, *], Anouk T. M. de Bakker[4, *], Heather. J. H. Johnstone[3] & Charlotte S. Miller[3]

[1]Department of Earth and Environmental Sciences, The University of Manchester, Williamson Building, Oxford Road, Manchester M13 9PL, United Kingdom.
[2]British Ocean Sediment COre Research Facility (BOSCORF), National Oceanography Centre, European Way, Southampton SO14 3ZH, United Kingdom
[3]Center for Marine Environmental Sciences (MARUM), University of Bremen, Klagenfurter Straße 4, 28359 Bremen, Germany.
[4]Unit of Marine and Coastal Systems, Deltares, Boussinesqweg 1, 2629 HV Delft, the Netherlands.
*These authors contributed equally: Diederik Liebrand, Anouk T. M. de Bakker.

*Correspondence to*: Diederik Liebrand (diederik@palaeoclimate.science) and/or Anouk T. M. de Bakker (anouk.debakker@deltares.nl)

**Abstract.** During the Late Pleistocene, Dansgaard-Oeschger (DO) cycles triggered warming events that were as abrupt as the present-day human-induced warming. However, in absence of a periodic forcing operating on millennial time scales, the main energy sources of DO cycles remain debated. Here, we identify the energy sources of DO cycles by applying a bispectral analysis to the North Greenland ice core project (NGRIP) oxygen isotope ($\delta^{18}O_{ice}$) record; a 123-thousand-years (kyr) long proxy-record of air-temperatures ($T_{air}$) over Greenland. For both modes of DO cyclicity—slow and fast—we detect disparate energy sources. Slow-DO cycles, marked by multi-millennial periodicities in the 12.5 to 2.5 kyr bandwidth, receive energy from astronomical periodicities. Fast-DO cycles, characterized by millennial periodicities in the $1.5 \pm 0.5$ kyr range, receive energy from centennial periodicities. We propose cryospheric and oceanic mechanisms that facilitate the transfer of energy from known sources to slow- and fast-DO cycles, respectively. Our findings stress the importance of understanding energy-transfer mechanisms across a broad range of time scales to explain the origins of climate cycles without primary periodic energy-sources.

## 1 Introduction

Climate variability on (multi-) millennial time scales is an enigmatic phenomenon because there is no consensus about the energy sources for this variability (Stuiver and Braziunas, 1993; Wara et al., 2000; Schulz et al., 2002; Braun et al., 2005; Sun et al., 2021; Zhang et al., 2021). Astronomical climate forcing, i.e., a nonlinear response of the Earth's climate system to changes in the distribution of incoming solar radiation (insolation) across the hemispheres/seasons, can explain the variability in palaeoclimate time series in the Milankovitch band (i.e., tens to hundreds of thousands of years) (Hays et al., 1976; Liebrand and de Bakker, 2019; Riechers et al., 2022). Similarly, solar forcing through modest changes in the total annual amount of insolation the Earth receives, resulting from the ~11-year sunspot cycle (±0.15%), can explain

palaeoclimate variability on time scales that range from the annual and decadal, up to the centennial (Stuiver and Braziunas, 1993; Wagner et al., 2001; Peristykh and Damon, 2003; Huybers and Curry, 2006). The presence of energy sources for climate variability on these time scales, starkly contrast with the absence of a widely accepted forcing agent on the in-between millennial and multi-millennial periodicities (a thousand to ten thousand years) (Pestiaux et al., 1988; Hagelberg et al., 1991; Hagelberg et al., 1994; Pelletier, 1998; Braun et al., 2005; Huybers and Curry, 2006; Braun et al., 2009; Scafetta et al., 2016; Rousseau et al., 2022). Insolation has been suggested to vary on these timescales (Scafetta et al., 2016; Kelsey, 2022), but the geological archive cannot be straightforwardly interpreted as a true recorder of (changes in the distribution of) solar output on millennial time scales, due to the many nonlinear response mechanisms of the Earth's climate system. To advance the debate about what "fuels" millennial climate cycles, it is thus essential to see if millennial climate variability is cyclic (i.e., periodic) (Ditlevsen et al., 2005), and if so, where the energy is derived from.

The most notable example of millennial climate variability are the DO cycles observed in the NGRIP $\delta^{18}O_{ice}$ record (Andersen et al., 2004). As an $T_{air}$ proxy record for the past 123 kyr (Late Pleistocene to Holocene), the NGRIP $\delta^{18}O_{ice}$ chronology is unsurpassed in its resolution, dating precision and accuracy, and detail of structure in the data (i.e., signal-to-noise ratio) (Andersen et al., 2004; Andersen et al., 2006; Rasmussen et al., 2006; Vinther et al., 2006; Svensson et al., 2008; Wolff et al., 2010; Kindler et al., 2014). This level of data quality may constitute a permanent ceiling in what can be reconstructed using climate proxies from any of Earth's natural archives, because no marine or terrestrial record will probably ever be able to match ice core chronologies in all these respects simultaneously. A comprehensive statistical analysis of the NGRIP $\delta^{18}O_{ice}$ record—including a higher order spectral analysis—is thus essential if we want to fully probe the climate dynamics across centennial, millennial, and astronomical time scales captured in this unique chronology. Previous statistical tests showed that certain spectral peaks in marine records were lower or higher harmonics of a primary cyclic signal (Hagelberg et al., 1994; Riechers et al., 2022); findings suggestive of frequency- and phase-coupling among climate cycles. However, these analyses were not able to detect direction and magnitude of energy transfers among climate cycles. We overcome the shortcomings of these methods by computing directions and magnitudes of energy transfers among climate cycles in the NGRIP $\delta^{18}O_{ice}$ record directly from the bispectrum; a statistical technique originally developed by the climatologist and 2021 Nobel-laureate in Physics Klaus Hasselmann and colleagues to investigate the nonlinear origins of near-shore ocean waves (Hasselmann et al., 1963). Our interpretation of the bispectral results is based on the premise that Earth's climate-cryosphere system behaves, at least partially, as a nonlinear oscillator (Le Treut and Ghil, 1983). We apply an advanced interpretation of the bispectrum to the NGRIP $\delta^{18}O_{ice}$ record (i.e., the total and zonal integrations of the imaginary part of the bispectrum) (Herbers et al., 2000; de Bakker et al., 2015; Liebrand and de Bakker, 2019) and show that both slow- and fast-DO cycles derive their energy from largely disparate astronomical and centennial sources through nonlinear interactions inherent to the climate-cryosphere system. These findings may abolish the need for invoking millennial forcing agents (Scafetta et al., 2016) to explain millennial climate cycles.

## 2 Asymmetric DO cycles in the NGRIP $\delta^{18}O_{ice}$ record

In total, 26 quasi-periodic DO cycles are currently recognized in the NGRIP $\delta^{18}O_{ice}$ record (see Methods), some of which are subdivided into a few shorter lasting episodes (Fig. 1a) (Dansgaard et al., 1993; Andersen et al., 2004; Rasmussen et al., 2014). They are characterized by abrupt warming events at the base of relatively warm interstadials ($\delta^{18}O_{ice}$ = –36 to –40 ‰,

$T_{air} \approx$ –35 to –40 ºC), followed by either several stepwise and rapid, or one more gradual, $T_{air}$ decrease(s), which lead(s) to longer-lasting and colder stadials ($\delta^{18}O_{ice}$ = –40 to –45 ‰, $T_{air} \approx$ –50 ºC) (Fig. 1a) (Kindler et al., 2014). For most DO events this evolution, i.e., an abrupt warming followed by a stepwise or a more gradual cooling, results in asymmetric (i.e., sawtooth-shaped) cycle shapes or waveforms (Hagelberg et al., 1991; King, 1996). According to bispectral theory, asymmetric cycle shapes in time series constitute a prognostic geometry for the nonlinear transfer of energy across the power

spectrum (Herbers et al., 2000; de Bakker et al., 2015). Outside of Greenland, climate variability highly comparable to the DO cycles in structure is recognized in planktonic foraminiferal $\delta^{18}O$ values from the Iberian Margin, indicating that sea-surface temperatures in the wider North Atlantic region were as variable on millennial time scales as the $T_{air}$ cycles over Greenland (Shackleton et al., 2000; Hodell et al., 2023). Evidence for the global climatic expression of Late Pleistocene millennial climate variability, albeit more subdued, comes from, e.g., benthic foraminiferal $\delta^{18}O$ values (i.e., deep water

temperatures) from the Iberian Margin (Shackleton et al., 2000; Hodell et al., 2023), from $\delta D$ (i.e., air-temperature) and methane records from the Antarctic ice core chronology (Loulergue et al., 2008), and from globally distributed speleothem records (Corrick et al., 2020; Menviel et al., 2020; Batchelor et al., 2023).

Prior to investigating the asymmetric cycle shapes of the NGRIP $\delta^{18}O_{ice}$ record in more detail, we reassess the (quasi-)

periodic nature of millennial climate cycles (Fig. 1c) (Schulz et al., 1999; Alley et al., 2001; Hinnov et al., 2002; Matyasovszky, 2010). We note that there is a lack of consensus in the literature about the presence/absence of periodic components in the NGRIP $\delta^{18}O_{ice}$ record, with some studies arguing against any form of periodicity but in favour of purely stochastically induced DO "events" (Ditlevsen et al., 2005; Ditlevsen et al., 2007; Kleppin et al., 2015; Gottwald, 2021). However, considering our new bispectral results that identify a clear grouping in the distribution of *nonsinusoidality* (i.e.,

asymmetry) per frequency (see next paragraph), we believe that also a reassessment of the distribution of *sinusoidality* per periodicity is justified. To do so, we apply a wavelet analysis to the data (Fig. 1c) (see Methods). A Gaussian notch-filter, applied to the NGRIP $\delta^{18}O_{ice}$ record prior to wavelet analysis, somewhat enhances the power of the millennial periodicities with respect to the astronomical periodicities. Similar to previous studies that performed spectral analysis on the NGRIP $\delta^{18}O_{ice}$ record (Petersen et al., 2013; Mitsui et al., 2019; Riechers et al., 2022), we identify two distinct modes of DO

variability in the time-periodicity domain: (*i*) slow-DO cycles, which are marked by sub-astronomical/multi-millennial periodicities in the 12.5 to 2.5 kyr bandwidth (e.g., DO cycles 1, 8, 12, 19, 20, and 22) (Dansgaard et al., 1984), and (*ii*) fast-DO cycles that have millennial periodicities in the 1.5 ± 0.5 kyr range (Dansgaard et al., 1984; Schulz et al., 1999; Schulz, 2002; Braun et al., 2005; Ditlevsen et al., 2007) that extends into the centennial periodicity range (e.g., DO cycles 3, 4, 5, 6,

7, 9, 10, 11, and 18) (Fig. 1c). Slow-DO cycles are near-continuously present throughout the record and are especially strongly expressed from 110 to105 thousand years ago (ka), from 80 to 70 ka, from 50 to 35 ka, and from 15 to 10 ka. Slow-DO cycles contribute to the variance associated with the Eemian and Holocene interglacials, as well as high amplitude DO Cycles 1 (i.e., Bølling-Allerød/Younger Dryas), 19 and 20 (Fig. 1). Fast-DO cycles are briefly present from 110 to 105 ka, and near-continuously expressed from 90 to 10 ka. The distinction in cycle durations between the two modes of DO variability suggests that there may be at least two distinct climatic mechanisms at their root-cause.

## 3 Energy transfers among climate cycles shed light on the "black box" climate system

To further probe the origins of both slow- and fast-DO cycles in the NGRIP $\delta^{18}O_{ice}$ record, we apply a bispectral analysis (Fig. 2) to the detrended, and Hamming-tapered data (see Methods). The Hamming-tapering focusses the time-averaged bispectral analysis on the central part of the 123 kyr long NGRIP $\delta^{18}O_{ice}$ record. Bispectral analysis is a statistical technique that can deconvolve the asymmetry (i.e., sawtoothness) of a periodic signal into three parts. These constituents, one sum-frequency ($f_3$, i.e., $f_3 = f_1 + f_2$) and two difference-frequencies ($f_1$ and $f_2$, i.e., $f_1 = f_3 - f_2$, and $f_2 = f_3 - f_1$), can either gain (orange and red colours for $f_3$, blue colours for $f_1$ and $f_2$) or lose (blue colours for $f_3$, orange and red for $f_1$ and $f_2$) energy (Fig. 2, Supp. Figs. S1, S2, Supp. Table S1). This makes bispectral analysis a powerful technique to identify energy sources of asymmetric periodic signals. In the bispectrum of the NGRIP $\delta^{18}O_{ice}$ record (Fig. 2), we observe three slow-DO periodicities that stand out by being involved in many energy exchanges, namely the 6.1, 4.2, and 3.7 kyr cycles. For fast-DO cycles, we observe more subtle interactions at the crossing lines with the 40 kyr obliquity and 23 kyr precession cycles, and, more prominently, with the 4.2 and 3.7 kyr slow-DO cycles. *Relative* energy transfers among climate cycles, and their directions, can thus be derived from so-called triad interactions displayed in (the imaginary part of) the bispectrum. The bispectrum shows that slow- and fast-DO cycles have highly complex, and mixed energy sources from across a wide range of periodicities. But, the *net* effect per periodicity of all individual energy exchanges cannot be derived from the bispectrum alone because individual periodicities are represented in the bispectrum as $f_1$, $f_2$, and $f_3$ simultaneously (i.e., $1/p_1$, $1/p_2$, and $1/p_3$), each of which can gain and/or lose energy (see Methods) (de Bakker et al., 2015; Liebrand and de Bakker, 2019).

To obtain magnitudes of net energy transfers per periodicity (i.e., "climate cycle") we need to integrate over the bispectrum (see Methods). We apply a coupling coefficient as part of this integration step to scale energy with frequency. The total integration of the bispectrum of the NGRIP $\delta^{18}O_{ice}$ record reveals three distinct bandwidths in the periodicity domain, each marked by distinct regimes in energy losses and/or gains (Fig. 3b). First, the astronomical periodicities of ~110 kyr eccentricity, 40 kyr obliquity, and ~20 kyr precession, lose energy. Second, periodicities of slow-DO cycles, covering the bandwidth from 12.5 to 2.5 kyr, predominantly gain net energy. Third, fast-DO cycles and centennial periodicities, in the 2.5 to 0.1 kyr periodicity range, are marked by rapidly alternating net energy gains and losses that seem to largely balance one another across this part of the spectrum; notwithstanding an overall loss in energy periodicities in the range between 0.3 and

0.1 kyr (Fig. 3b). The net loss of energy at the ~110 kyr eccentricity periodicity is a surprising result that contrasts with a Milankovitch Theory-based understanding of precession and obliquity fuelling of eccentricity variability of Earth's climate-cryosphere system (Liebrand and de Bakker, 2019); a result that should be interpreted with caution (see Methods). Net energy gains that stand out in amplitude are the slow-DO cycles with periodicities of 6.1, 4.2, and 3.7 kyr, as well as the gains at the 2.9, and 2.7 kyr periodicities. Some further notable net energy gains coincide with the 1.5 ± 0.5 kyr periodicity range of the fast-DO cycles, such as the 1.7, 1.5, and 1.3 kyr periodicities. Other well-expressed gains fall in the centennial bandwidth; being the 0.9, 0.5, 0.4, and 0.2 kyr periodicities. By integrating over the bispectrum we show which periodicities gain and lose energy, and thus how energy is redistributed over the power spectrum. Based on this new information we can piece together cause-and-effect relationships otherwise unobservable within the "black-box" of Earth's climate system and derive an origin-story for both slow- and fast-DO cycles.

By comparing the power spectrum of insolation (Fig. 3a) to the total integration of the bispectrum (Fig. 3b) and power spectrum (Fig. 3c) of the NGRIP $\delta^{18}O_{ice}$ record, we show that part of the energy the Earth receives on astronomical time scales (Fig. 3a) is lost (Fig. 3b) and transferred toward shorter periodicities, particularly those of slow-DO cycles (Fig. 3b, c). Slow-DO cycles thus have an indirect astronomical origin (Hagelberg et al., 1994; Wara et al., 2000; Rial and Saha, 2011; Sun et al., 2021; Zhang et al., 2021), at least in part. Furthermore, we observe several distinct energy gains in the 1.5 ± 0.5 kyr periodicity range of fast-DO cycles, which are accompanied by smaller amplitude energy losses. These net gains show that part of the spectral power associated with fast-DO cycles is derived from nonlinear interactions with other climate cycles (Fig. 3b, c). However, from the total integration of the bispectrum, it is hard to make out whether most of this energy is derived from astronomical or other fast-DO and centennial cycles.

To better identify the disparate energy sources for both slow- and fast-DO cycles, we devise a zonation scheme (Fig, 2). This zonation scheme aims to isolate groups of nonlinear interactions among specific frequency bandwidths. It consists of nine unique combinations of the three main bandwidths: (*i*) astronomical, (*ii*) slow-, and (*iii*) fast-DO and centennial periodicities (Fig. 2, Supp. Fig. S1, Supp. Table S1). Boundaries for the zonal integration are selected at frequencies where the qualitative behaviour of energy transfers in the total integration changes (Fig. 3b). This results in $f$ = 80 Myr$^{-1}$ ($p$ = 12.5 kyr) for the boundary between astronomical and slow-DO cycles, and at $f$ = 400 Myr$^{-1}$ ($p$ = 2.5 kyr) for the boundary between slow-DO and fast-DO cycles. Subsequently, we integrate the bispectrum of the NGRIP $\delta^{18}O_{ice}$ record zonally (Fig. 4) (see Methods) (de Bakker et al., 2015; Liebrand and de Bakker, 2019). From the zonal integrations we learn that most energy is exchanged among astronomical and slow-DO cycles (Zone 3, Fig. 4c), which results in a net transfer of energy to the 3.7 kyr cycle. Other energy sources for slow-DO cycle now also become apparent; additional energy is sourced from fast-DO periodicities in the range between 2.0 and 2.5 kyr (Zone 6, Fig. 4f) and broadly from across the millennial to centennial periodicity range (Zone 8, Fig. 4h). Fast-DO cycles, on the contrary, do not appear to receive much energy from astronomical periodicities

(Zone 7, Fig. 4g), but are "fuelled" by a combination of centennial climate cycles and secondary interactions with other fast-DO periodicities (Zones 6-9, Fig. 4f-i).

## 4. Potential cryospheric, climatic, and oceanic energy-transfer mechanisms

By statistically reanalysing the NGRIP $\delta^{18}O_{ice}$ record, we find clear confirmatory (Petersen et al., 2013; Vettoretti and Peltier, 2018) evidence for two distinct modes (i.e., slow, and fast) of DO variability (Fig. 1c and Fig. 3b). Furthermore, based on our advanced bispectral analyses (Figs. 2, 3b, and 4), we show that disparate fuelling pathways underpin these two modes of DO cyclicity. If these interpretations are correct, then slow-DO cycles constitute a nonlinear response primarily to astronomical climate forcing (Hagelberg et al., 1994; Wara et al., 2000; Rial and Saha, 2011; Sun et al., 2021; Zhang et al., 2021), which in combination with secondary energy-redistributing mechanisms among slow-DO periodicities results in $T_{air}$ cyclicity over Greenland in the 12.5 to 2.5 kyr bandwidth. It is important to note that the direction of energy transfers, from obliquity and precession periodicities (and potentially eccentricity periodicities, see Methods) to the multi-millennial slow-DO periodicities (Figs. 2, 3b, and 4), is inversed (i.e., from longer to shorter periodicities) from what has been documented for the Middle and Late Pleistocene 40-kyr and ~110-kyr worlds (Liebrand and de Bakker, 2019). These ice age cycles are, in accordance with the Milankovitch Theory, marked by energy transfers from precession to obliquity periodicities and from precession and obliquity to eccentricity periodicities, respectively (i.e., from shorter to longer periodicities). We speculate that the energy-redistributing mechanisms from precession and obliquity periodicities to and among slow-DO periodicities were predominantly linked to the Northern Hemisphere climate-cryosphere system (Vettoretti and Peltier, 2018). For example the regeneration time, or waiting time (Alley et al., 2001), of grounded ice in the Hudson Bay following catastrophic collapses (Boers et al., 2018) could be astronomically forced. In such a scenario, the rate of slow-DO cooling phases and grounded ice expansion (i.e., inceptions), potentially in combination with floating ice shelf expansions near Greenland, were paced by insolation conditions. Yet their warming phases (i.e., terminations), would have been triggered by the passing of (an) internal threshold(s), most probably during the NH summer. Such a mechanism, somewhat comparable to the binge/purge model for continental-sized ice sheets (MacAyeal, 1993), would result in asymmetric cycle shapes that mark the slow-DO cycles, as well as have a short-periodic nonlinear response to insolation forcing built in, as we document in our bispectral results. These quasi-regular oscillations would have affected sea-ice cover and regional $T_{air}$, most probably also over Greenland. Furthermore, the relatively long time periods needed for ice sheets to respond, are in line with a relatively slow mode of DO cyclicity.

For fast-DO cycles, the energy redistributing mechanisms (i.e., from shorter to longer periodicities) are not as straightforward as those for slow-DO cycles. However, our results suggest that many, closely spaced energy exchanges in the frequency domain (see e.g., the rapidly alternating losses and gains in Fig. 3b), may have served as a primary energy source for fast-DO cycles in the 1.5 ± 0.5 kyr bandwidth. Although energy for fast-DO cycles may have been collected from

many centennial periodicities (Ditlevsen and Crucifix, 2017), mainly in the 0.3 to 0.1 kyr range (Fig. 3b and 4), initially this energy was probably derived from long-term modulations in the solar cycles (Wagner et al., 2001; Peristykh and Damon, 2003), or derived from even shorter-periodic (e.g., decadal) sunspot cycles not captured in our analysis. We tentatively support a solar forcing hypothesis for fast-DO cycles, albeit not through a summing of periodicities (Braun et al., 2005), but through nonlinear energy exchanges (i.e., triad interactions) with shorter periodicities. We speculate that changes in ocean currents linked to Atlantic meridional overturning circulation (AMOC) and/or tides, may be the primary driver for the distinct fast-DO mode of variability (Dokken et al., 2013; Vettoretti and Peltier, 2018; Griem et al., 2019; Armstrong et al., 2022). In this scenario, AMOC strength had destabilizing effects on floating ice shelfs only, but not affecting grounded ice as much. Ice shelf collapse and rapid warming followed by a somewhat slower phase of ice shelf re-expansion, would explain the asymmetric shape of the fast-DO cycles (Schulz et al., 1999). Both modes of DO variability, slow and fast, were likely amplified through positive feedbacks of the carbon cycle (Brook et al., 1996; Bauska et al., 2021; Vettoretti et al., 2022). In fact, by chemically storing energy at certain time scales/periodicities, and releasing it at others, the carbon cycle itself could have functioned as an energy-redistributing mechanism. A rechargeable battery may serve as a metaphor for such a process. It is thus important to consider both physical (e.g., build up and loss of grounded and floating ice) and chemical (e.g., energy stored in, and released from, organic carbon) energy sinks and sources for the nonlinear redistribution of energy across time scales, as quantified in the bispectrum.

## 5 Conclusions

DO cycles—both slow and fast—probably resulted, at least in part, from cryospheric, climatic, and potentially oceanic energy-redistributing processes that operated on a broad continuum of periodicities and linked dynamic behaviour across time scales (Schulz et al., 1999; Schulz, 2002; Schulz et al., 2002; Boers et al., 2018; Armstrong et al., 2022; Vettoretti et al., 2022). We provide a description of two distinct energy transfer pathways: one from longer astronomical periodicities, and one from shorter centennial periodicities to the (multi-) millennial periodicities that mark the DO cycles. We emphasise, that the disparate energy sources for DO variability that we identify in the bispectrum of the NGRIP $\delta^{18}O_{ice}$ record, constitute periodic Earth-internal sources of energy. However, both these Earth-internal sources are, in turn, externally forced by insolation variability on either astronomical or centennial (and shorter) timescales. The disparate energy sources and associated climate mechanisms contrast with previous studies that favour a single mechanism to explain both slow- and fast-DO cyclicity (e.g., (Petersen et al., 2013; Lohmann and Svensson, 2022; Vettoretti et al., 2022)). The bispectrum can only assess net energy exchanges among periodic components of a time series, and hence, we cannot rule out non-periodic, non-energy redistributing forcing factors, such as volcanism (Crick et al., 2021; Paine et al., 2021; Lin et al., 2022; Lohmann and Svensson, 2022), asteroid impacts (van Hoesel et al., 2014), and/or noise induced or other "stochastic events" (Alley et al., 2001; Braun et al., 2009; Lohmann et al., 2020; Vettoretti et al., 2022). However, our study stresses the importance of

considering nonlinear—but deterministic—energy-redistributing mechanisms within the climate-cryosphere system to explain periodic signals on time scales without primary periodic energy-sources.

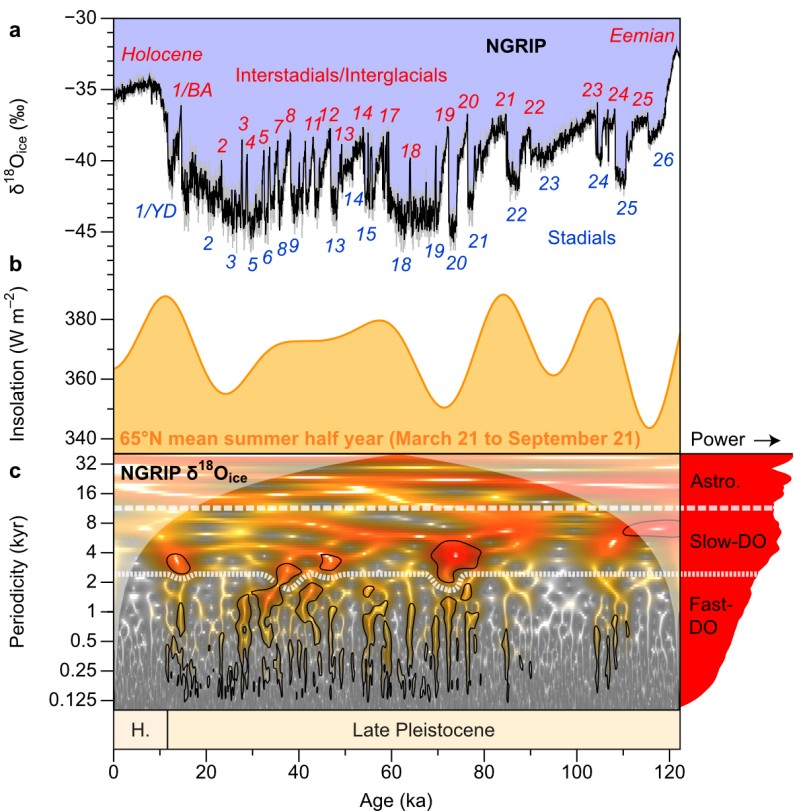

**Figure 1: The NGRIP δ¹⁸O$_{ice}$ record versus summer insolation at 65° N. (a)** The NGRIP δ¹⁸O$_{ice}$ record on the "GICC05modelext" age model (Andersen et al., 2006; Rasmussen et al., 2006; Vinther et al., 2006; Svensson et al., 2008; Wolff et al., 2010) versus **(b)** mean summer half year insolation at 65° N (March 21 to September 21) for the past 123 ka (Laskar et al., 2004). In panel **(a)**: grey line represents the NGRIP δ¹⁸O$_{ice}$ record with 20-year sample spacing, used for

bispectral analyses (Fig. 2-4). Black line represents the NGRIP δ¹⁸O$_{ice}$ record with a 50-year sample spacing, used for the wavelet analysis (panel **c**). **(c)** Wavelet analysis of the Gaussian filtered NGRIP δ¹⁸O$_{ice}$ record (see Methods). Black contours represent the 95% significance level with respect to a red noise model. The colours of the wavelet indicate amplitude (white/grey = low amplitude, yellow/orange/red = high amplitude). We note that colour/amplitude does not equate to statistical significance with respect to the red noise model. An average of the entire wavelet spectrum is shown on the right-

hand side in red. The two white dashed lines indicate the spectral gaps between astronomical and slow-DO periodicities (broad dashes), and between slow-DO and fast-DO periodicities (narrow dashes). "H" stands for Holocene, "BA" stands for Bølling-Allerød, and "YD" stands for Younger Dryas. The latter two together correspond to DO Cycle-1.

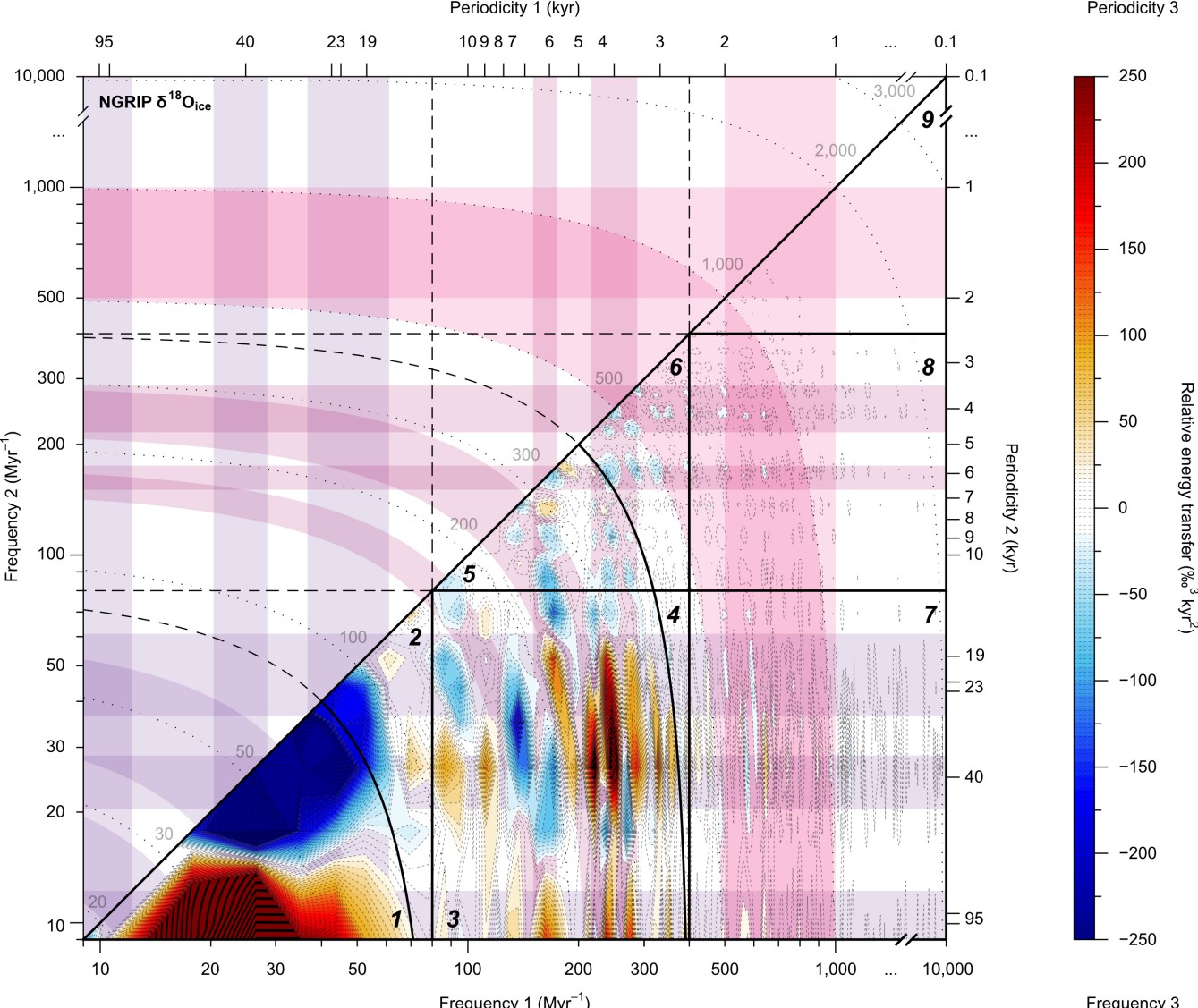

**Figure 2: Bispectrum of the NGRIP δ¹⁸Oice record.** The imaginary part of the bispectrum of the NGRIP δ¹⁸Oice record depicts energy exchanges among three frequencies. Energy can only be transferred if both frequencies- ($f$) and phases- ($\varphi$) are coupled: $f_1 + f_2 = f_3$ and $\varphi_1 + \varphi_2 = \varphi_3$. The x-axis shows difference-frequency $f_1$, the y-axis depicts difference-frequency $f_2$, the colour bar corresponds to sum-frequency $f_3$ (i.e., $f_1 + f_2$). Two outcomes exist for energy exchanges: either $f_3$ gains energy and $f_1$ and $f_2$ lose energy, which is shown in orange and red colours, or $f_3$ loses energy and $f_1$ and $f_2$ gain energy, which is shown in blue colours. Purple, plum, and pink coloured bands mark the main periodicities for astronomical, slow-, and fast-DO cycles, respectively. Vertical and horizontal coloured-bands mark $f_1$ and $f_2$, respectively. Diagonal bands (curved on the log-log scale) mark $f_3$. Thick solid black lines mark our selected zone boundaries between astronomical, slow-, and fast-DO

and centennial periodicities. See Supp. Fig. S2 for a linear-linear scale version of this figure. For a more detailed explanation of bispectral analysis we refer to the Methods and Liebrand and de Bakker (2019).

255

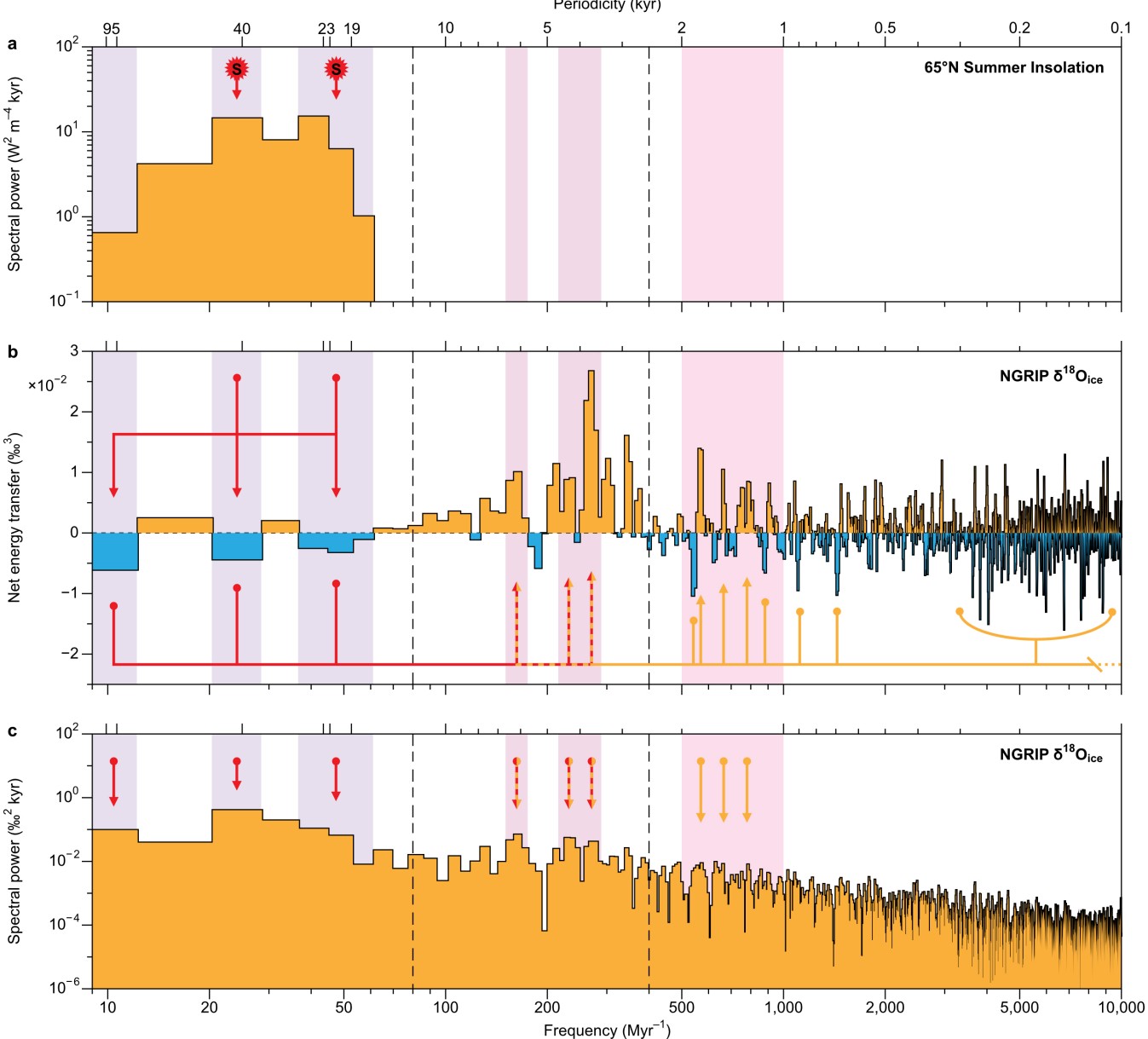

**Figure 3: Linking DO cycles to energy sources.** Comparison of (**a**) power spectrum of mean summer half year insolation record at 65ºN (March 21 to September 21), to (**b**) total integration of the bispectrum of the NGRIP $\delta^{18}O_{ice}$ record, and (**c**) the power spectrum of the NGRIP $\delta^{18}O_{ice}$ record. Orange indicates spectral power (panel **a** and **c**), or net energy gains (panel **b**), and blue indicates net energy losses (panel **b**). Red and orange arrows indicate the two main energy transfer pathways

across the spectral continuum to slow- and fast-DO cycles. These interpretative arrows are derived from both the total (this figure) and the zonal integration (see also Fig. 4). Dashed vertical black lines mark the zone boundaries between astronomical periodicities (purple), slow- (plum), and fast-DO (pink) and centennial periodicities. "S" stands for sun/solar forcing, indicating at which frequencies summer insolation provides energy. No 123 kyr long record of sunspot cycles exist, which is why we do not show a spectrum of this energy source here. Furthermore, the highest amplitude spectral power of the solar cycle over the past 123 kyr is expected to fall at around the annual and ~11-year periodicities. These cycles are not resolved in the NGRIP $\delta^{18}O_{ice}$ record, which is marked by a 20-year sample spacing.

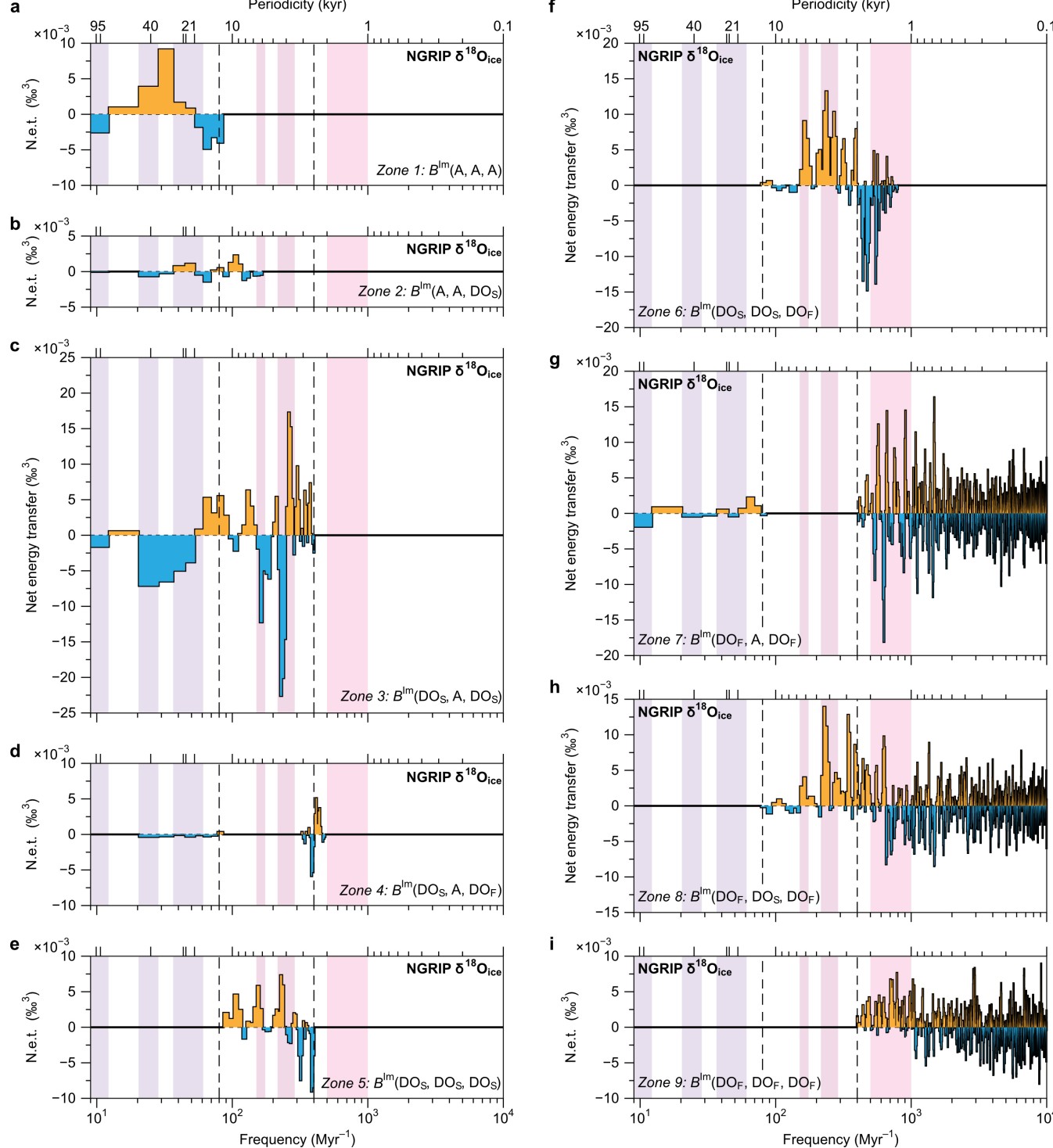

**Figure 4: Zonal integrations of the bispectrum.** Integrations of (**a**) Zone 1: $B^{Im}(A, A, A)$, (**b**) Zone 2: $B^{Im}(A, A, DO_S)$, (**c**) Zone 3: $B^{Im}(DO_S, A, DO_S)$, (**d**) Zone 4: $B^{Im}(DO_S, A, DO_F)$, (**e**) Zone 5: $B^{Im}(DO_S, DO_S, DO_S)$, (**f**) Zone 6: $B^{Im}(DO_S, DO_S,$

DO$_F$), (**g**) Zone 7: $B^{Im}$(DO$_F$, A, DO$_F$), (**h**) Zone 8: $B^{Im}$(DO$_F$, DO$_S$, DO$_F$), and (**i**) Zone 9: $B^{Im}$(DO$_F$, DO$_F$, DO$_F$). "$B^{Im}$" stands for the imaginary part of the bispectrum. "A" stands for astronomical periodicities, "DO$_S$" stands for slow-DO periodicities, "DO$_F$" stands for fast-DO and centennial periodicities. Orange indicates net energy gains, blue net energy losses. Dashed vertical black lines mark the zone boundaries between the main periodicity bandwidths, with the main astronomical periodicities highlighted in purple, slow-DO periodicities in plum, and fast-DO periodicities in pink.

## Methods

**NGRIP age model.** The NGRIP record is composed of two bore holes that are spliced together. These cores recovered snow and ice layers with annual resolution. To generate a high-fidelity time series 20 layers were averaged, yielding a highly reproducible and largely noise-free δ$^{18}$O$_{ice}$ chronology. The ice-water δ$^{18}$O record constitutes an T$_{air}$ proxy record of the past 123 kyr that captures astronomical, millennial, and centennial climate variability in exquisite detail (Andersen et al., 2004; Kindler et al., 2014) (Fig. 1). The "GICC05modelext" age model (Andersen et al., 2006; Rasmussen et al., 2006; Vinther et al., 2006; Svensson et al., 2008; Wolff et al., 2010) used here, was constructed using a combination of annual ice layer counting and an ice flow model. We note that the relatively small age model differences (up to ~0.5 kyr) compared to the older "ss09sea" age model (Andersen et al., 2004) do not affect the outcome of bispectral analyses, because bispectra assess cycle geometry in the frequency domain, which allows for modest changes in the time domain (Liebrand and de Bakker, 2019).

**Statistical analyses.** The wavelet analysis of the NGRIP δ$^{18}$O$_{ice}$ record transforms the time series into the time-periodicity domain using a Morlet wavelet (Fig. 1c). Prior to the wavelet analysis only, we applied a Gaussian notch filter ($f = 0$ kyr$^{-1}$, bandwidth = 0.05 kyr$^{-1}$, (Paillard et al., 1996)) to the NGRIP δ$^{18}$O$_{ice}$ record (50-year sample spacing) to somewhat enhance the power of the millennial periodicities with respect to the astronomical periodicities (i.e., the latter are suppressed by the notch filter).

In this study, we use the same bispectral methods as described in Liebrand and de Bakker (2019). In short: the bispectrum is defined by Eq. (1):

$$B_{f_1, f_2} = E[C_{f_1} \, C_{f_2} \, C^*_{f_1 + f_2}], \tag{1}$$

where $E[\ldots]$ is the ensemble average of the triple product of complex Fourier coefficients $C$ at the difference frequencies $f_1$, $f_2$, and their sum $f_3$ (i.e., $= f_1 + f_2$), and the asterisk indicates complex conjugation (Hasselmann et al., 1963). Energy gains and losses in both the bispectrum and the total and zonal integration of the bispectrum are inverted compared to Liebrand and de Bakker (2019) because the δ$^{18}$O$_{ice}$ records an inverted signal compared to benthic foraminiferal δ$^{18}$O. To extract the total

gain or loss in energy per climate cycle over the 123 kyr span of the NGRIP $\delta^{18}O_{ice}$ record, and hence, to obtain conservative

net energy transfers per frequency (defined by the nonlinear source term $S_{nl}$), we integrate over the imaginary part of the bispectrum and multiply it with a coupling coefficient. Similar to Liebrand and de Bakker (2019), we make minimum assumptions and use a coupling coefficient that only corrects for frequency $W_{(f_1,f_2)} = (f_1 + f_2)$, which is also part of the Boussinesq scaling used for ocean waves (Herbers and Burton, 1997; Herbers et al., 2000). The correction for frequency ($W_{(f_1,f_2)}$) ensures energy conservation during triad interactions (see Supp. Fig. S3) and permits qualitative interpretations of

conservative net energy exchanges across the spectrum (see Fig. 3b and Fig. 4). We express the integral of $S_{nl}$ in terms of an integration, or summation, over the positive quadrant of the bispectrum alone, or equivalently in sum and difference interactions following Eq. (2):

$$S_{nl,f} = S_{nl,f+} + S_{nl,f-},\qquad\qquad(2).$$

The difference contributions are expressed by Eq. (3):

$$S_{nl-} = -2\sum_{f\in F-} W_{f+f',-f'}\, I\{B_{f,f'}\}.\qquad\qquad(3)$$

and are obtained by integrating along vertical and horizontal lines, perpendicular to the $x$ and $y$-axes, respectively (Fig. 2, Supp. Fig. S2). The sum contributions are expressed by Eq. (4):

$$S_{nl+} = \sum_{f\in F+} W_{f',f-f'}\, I\{B_{f',f-f'}\},\qquad\qquad(4)$$

and are obtained by integrating diagonally over the bispectrum (Fig. 2, Supp. Fig. S2). Integration can be performed including all frequencies, i.e., a total integration over the entire imaginary part of the bispectrum (Fig. 3b), or over specific zones only, in case subsets of interactions need to be further examined (Fig. 4, Supp. Table S1) (de Bakker et al., 2015; de Bakker et al., 2016).

The imaginary part of the bispectrum, the total, and the zonal integrations, constitute an average of the past 123 ka. Prior to bispectral analysis, the raw NGRIP $\delta^{18}O_{ice}$ record (20-year sample spacing) was linearly detrended and multiplied by a Hamming window function. We subsequently used an energy correction factor that corrects for the Hamming tapering. Previous statistical analyses (Liebrand and de Bakker, 2019) have shown that this approach considerably clarifies the bispectral results. Resulting from the tapering, nonlinear triad interactions in the central part of the time-series are relatively

over-represented, whereas those on either end of the record (i.e., the Eemian, Bølling-Allerød/Younger Dryas, and Holocene) are relatively underrepresented in the bispectral analysis. Furthermore, only one ~110 kyr cycle is present in the

123-kyr long NGRIP $\delta^{18}O_{ice}$ record. The structure of this ~110-kyr cycle, and hence the nonlinear triad interactions in the bispectrum, are affected by the Hamming tapering. We thus caution against overinterpreting the energy loss observed at the ~110-kyr eccentricity periodicity (i.e., the very lowest frequency), which is at, or beyond, the detection limit of our analysis. Future analyses on longer records, marked by both astronomical and millennial climate variability, are needed to verify the robustness of the energy loss at the ~110-kyr eccentricity periodicity in the NGRIP $\delta^{18}O_{ice}$ record, because longer window lengths, that incorporate multiple ~110-kyr cycles, can then be selected (see also Liebrand and de Bakker (2019)). We refer the interested reader to Schmidt (2020) for some of the latest applications and interpretations of bispectral analysis.

Following studies on ocean waves (Herbers et al., 2000; de Bakker et al., 2015), we focus the interpretation on the imaginary part of the bispectrum (Figs. 2, 3b, and 4), which is related to cycle asymmetry, because most energy transfers are associated with asymmetric cycle shapes/wave forms. Furthermore, we visually observe sawtoothshaped/asymmetric climate cycles in the NGRIP $\delta^{18}O_{ice}$ record, associated with both slow- and fast-DO cycles. It is key to note that the asymmetry/sawtoothness of DO cycles, analysed here using the imaginary part of the bispectrum, is distinct from their skewness/peakedness (Ditlevsen et al., 2005), which is deconvolved in the real part of the bispectrum, or from excess kurtosis/rectangularity/square-waveness (Ditlevsen et al., 1996; Hinnov et al., 2002), which is deconvolved in the trispectrum. The latter two constitute wholly different nonsinusoidal cycle properties (Hagelberg et al., 1991; King, 1996). Skewed and kurtose cycle geometries are not as straightforwardly associable with energy transfers (Herbers et al., 2000; de Bakker et al., 2015; Liebrand and de Bakker, 2019).

The bispectrum yields nonconservative relative energy transfers that have not yet been corrected for frequency (de Bakker et al., 2015; Liebrand and de Bakker, 2019). Prior to such a scaling, the relative energy transfers at higher frequencies are not as pronounced as those at lower frequencies in the bispectrum (Fig. 2). During the integration of the bispectrum, we scale energy transfers to frequency by assuming a standard coupling coefficient, which is based on the second order Boussinesq theory (Herbers and Burton, 1997; Herbers et al., 2000). This scaling yields conservative net energy transfers for the imaginary part of the bispectrum only (de Bakker et al., 2015; Liebrand and de Bakker, 2019) (Fig. 3, 4, and Supp. Fig. S3). We note, however, that both the nonconservative relative energy transfers of the bispectrum, as well as the conservative net energy transfers of the integration of the bispectrum, are not absolute (Liebrand and de Bakker, 2019). We forego scaling energy transfers to insolation due to the many unknowns in the climate-cryosphere system that link changes in $\delta^{18}O_{ice}$ (i.e., in ‰) to changes in insolation (i.e., in W m$^{-2}$).

## Code availability

The detailed mathematical description of the bispectral analyses used in this study are presented in the Methods as well as in de Bakker et al., (2015) and Liebrand and de Bakker (2019). Code will be made available on individual request to the corresponding authors.

## Data availability

The NGRIP $\delta^{18}O_{ice}$ record can be downloaded here: https://www.iceandclimate.nbi.ku.dk/data/2010-11-19_GICC05modelext_for_NGRIP.xls

## Author contributions

D.L. and A.T.M.d.B. designed the study, performed the data analysis, and made the figures. A.T.M.d.B. adjusted the bispectral MATLAB scripts for the purposes of this study. All authors discussed the interpretation of the results and contributed to the writing of the manuscript.

## Competing interests

The authors declare no competing interests.

## Acknowledgements

This project was funded in part by (*i*) the Deutsche Forschungsgemeinschaft (DFG) through a Cluster of Excellence grant awarded to MARUM, and (*ii*) the Natural Environment Research Council (NERC) through funding awarded to the British Ocean Sediment COre Research Facility (BOSCORF) and (*iii*) through NERC grant #NE/X002519/1 awarded to Dr. Neil C. Mitchell. Publication costs were paid by the University of Bremen/MARUM. We thank the NGRIP members and the wider ice-core drilling and science community for making the $\delta^{18}O_{ice}$ data and accompanying age models available. We thank T. H. C. Herbers and B. G. Ruessink for making their original bispectral MATLAB scripts available, which were adapted for this study. We thank A. Yool for help with the three-dimensional visualization of the wavelet analysis. We thank David de Vleeschouwer and an anonymous referee for their constructive feedback. We thank the anonymous referees that helped improve several earlier versions of this manuscript (first submitted in April 2022). We thank the handling editor Alexey Ekaykin and the Climate of the Past team for publishing this manuscript.

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
