# Peer review of "Disparate energy sources for slow and fast Dansgaard-Oeschger cycles"

_Climate of the Past, 2023_

## Author Response (AR1)

**Peer review „Disparate energy sources for slow and fast Dansgaard-Oeschger cycles"**

*We are thankful to David De Vleeschouwer for reviewing our manuscript. We have found his critical review and suggestions to improve and clarify the manuscript very helpful.*

Our climate system is full of rhythmical features on a wide variety of time-scales: tidal cycles, the daily cycle, the yearly cycle, sunspot cycles on decadal and centennial timescales, and Milankovic forcing on timescales of several ten thousands of years, ... For most of the rhythmical features in our climate system, we have a straight-forward explanation and the driving mechanism is typically astronomical in nature. However, for some cyclicity observed in the Earth climate history, the explanation is not that straight-forward. The most notable example is the 100-kyr problem, referring to the discrepancy between the 100-kyr rhythm in late Pleistocene glacial/interglacial cycles and the absence of a 100-kyr cycle in the insolation forcing. The Dansgaard-Oeschger cycles constitute a similar problem: The last glacial period was characterized by major climate variability on millennial timescales (stadial/interstadial variability), yet there is no obvious forcing mechanism at hand. To provide deeper insight into nature of this cyclicity, the authors apply bispectral analysis to the North Greenland ice core project (NGRIP) $\delta^{18}O_{ice}$ record. They conclude that "slow" Dansgaard-Oeschger cycles (3.5 – 7 kyr periods) are driven by the energy-transfer from astronomical frequencies. They conclude that "fast" Dansgaard-Oeschger cycles (1 – 2 kyr periods) are driven by the energy-transfer from centennial sunspot cycles (e.g., the 200-year de Vries cycle). In other words, the authors provide evidence for an external forcing for the Dansgaard-Oeschger cycles.

After reviewing the manuscript, I very much appreciate the novelty of the statistical approach the authors have applied. I commend them for explaining the approach in a clear and concise manner to an audience that is not yet very familiar with bispectral analysis. The figures presented in the manuscript require some effort to fully understand. Yet, they are well designed, easy to read and convincingly illustrate the claims made by the authors. Furthermore, I believe this work has the potential to become an important stepstone towards a more comprehensive understanding of Dansgaard-Oescher events, which have been the subject of long-standing debates (internal vs. external forcing). The authors' fresh and original view provides new insights into this complex phenomenon and will be of interest to the Quaternary paleoclimate community.

That being said, I have two major comments/questions that need to be addressed before the manuscript can be accepted for publication.

1. **The 100-kyr eccentricity cycle is presented as a net source of energy for "slow" Dansgaard-Oeschger cycles (Fig. 3b). However, the 100-kyr glacial-interglacial cycle itself only exists by the energy-transfer of precession and obliquity to the 100-kyr band**. It is because of this major comment that I started my review by making the analogy between the 100-kyr problem and the Dansgaard-Oeschger cycles. Most paleoclimate researchers now agree that the 100-kyr glacial-interglacial cycle is the result of a non-linear response of the Earth System to high-latitude insolation forcing. Thereby, an energy transfer occurs from the precession band towards the 100-kyr eccentricity band. Hence, I am surprised that this energy-transfer does not show up in Fig 2. To the contrary, the lowermost purple horizontal band in Figure 2 is entirely showing up in orange and red colors. This implies that the bispectral analysis is suggesting that the 100-kyr cycles ($f_2$ in this case on Fig. 2) is providing the energy to "fuel" climate variability in the obliquity band, the precession

band, and the "slow" D-O band. The same observation is illustrated in Fig. 4a (100-kyr transferring energy to obliquity/precession) and Fig. 4c (astronomical frequencies including the 100-kyr cycle transferring energy to the "slow" D-O cycles). Can the authors comment on this -to -me- surprising outcome of the bispectral analysis?

*This is indeed an unexpected finding in the bispectrum. We note, however, that (i) the entire NGRIP $\delta^{18}O$ record is 123 kyr long, or about the length of one single ~100 kyr cycle, and that (ii) we apply a Hamming tapering function that focusses the bispectral analysis on the central part of the 123 kyr long record, thereby altering the structure (i.e., asymmetric shape) of the longest periodicity, in this case the ~100 kyr. The ~100 kyr periodicity is thus on, or beyond, the detection limit of our analysis, and its apparent role in fuelling the obliquity, precession, and slow-DO cycles remains questionable and uncertain for now.*

*We will add a brief discussion to the manuscript that explains the caution that should be taken when interpreting interactions with the ~100 kyr cycle of the NGRIP $\delta^{18}O$ record.*

*As an aside, we would like to draw attention to Liebrand and de Bakker (Climate of the Past, 2019) who applied bispectral analysis to the "LR04" globally integrated benthic foraminiferal oxygen isotope stack (Lisiecki and Raymo, Paleoceanography, 2005). Unlike the NGRIP $\delta^{18}O$ record, the LR04 record is not limited to a time span of 123 kyr. Therefore, a longer window size of 668 kyr could be selected that was better suited to quantify energy exchanges among astronomical periodicities, including eccentricity paced periodicities. In Liebrand and de Bakker (2019), a much more robust precession (i.e., mainly low latitude) and obliquity (i.e., high latitude) "fuelling" of ~100 kyr ice age cycles was detected in the post-Mid Pleistocene Transition interval, in good agreement with the Milankovitch Theory.*

2. I agree with the authors that the sawtooth-shaped D-O cycles are an indication for the nonlinear transfer of energy across the power spectrum. Again, very similar to the 100-kyr cycles. **However, I would like to read a little bit more details on potential mechanisms facilitating this energy transfer.** The authors mention the regeneration time of grounded ice in the Hudson Bay to explain the slow cooling during a D-O cycle, and link the catastrophic collapse of this ice volume with the abrupt warming. They emphasize that the asymmetrical build-up and collapse of the cryosphere would have also influenced ocean circulation (lines 160 – 167). While I agree that the ocean and the cryosphere are likely at the origin of the nonlinear climate behavior, I don't consider this paragraph as a sufficient and satisfactory explanation for this paper. I do not understand how energy can be transferred from astronomical frequencies to the slow D-O band via these processes. I miss some kind of conceptual model. I am explicitly thinking about the Imbrie and Imbrie model for the 100-kyr cycles, quantitatively explaining how the asymmetry between ice buildup and ice melt causes a transfer of energy from the high frequencies to the low frequencies. The authors propose an energy transfer in the reverse direction, from low astronomical frequencies to higher D-O frequencies. To me, such a low-to-high transfer seems more difficult to explain... This is why I would encourage the authors to provide somewhat more details on possible mechanistic pathways that could explain this transfer. Even when such a discussion may be rather speculative, I think this is important to make this paper a completer and more comprehensive story.

*We agree that a better mechanistic understanding of how energy is transferred among periodic components of the climate-cryosphere system is desirable. In the bispectrum, frequency-frequency interactions can result in energy transfers to both lower and higher frequencies. From high frequencies (obliquity and precession) to low frequencies, is indeed explained by the Milankovitch Theory (see also Liebrand and de Bakker, 2019), which proposes that the winters are always cold enough in the high northern latitudes (regardless of orbital configuration), but that summer insolation only sometimes triggers ice melt (i.e., during a precession minimum and/or an obliquity maximum, corresponding to a high northern latitude insolation maximum).*

*We also agree that in this study, for slow-DO cycles, we aim to speculate on a mechanistic explanation for energy transfers in the other direction; from the relatively low, precession and obliquity frequencies to higher, slow-DO frequencies. We admit that our current explanation does not address this issue satisfactorily and will amend this paragraph (lines 160-167) in the revised version of this paper.*

*We will seek an explanation in which the rate of DO cooling phases (potentially linked to ice shelf expansions near Greenland) are paced by insolation conditions (i.e., orbital configurations related to both obliquity and precession), yet their warming phases (terminations) may be triggered by the passing of (an) internal threshold(s) during the NH summer. For example, Hudson Bay is at full ice shelf carrying-capacity, and once it starts shedding its ice shelf, it disintegrates ("retreats") completely. Such a mechanism/ conceptual model, like the binge/purge model for ice sheets, would result in asymmetric cycle shapes that mark the slow-DO cycles, as well as have a high-frequency/short-periodic nonlinear response to insolation forcing built in, as we document in our bispectral analysis.*

Overall, this manuscript has the potential to make an important contribution to the field and with the necessary revisions, could be a valuable addition to the literature.

*References cited:*

*Liebrand, D., & de Bakker, A. T. M., Bispectra of climate cycles show how ice ages are fuelled, Climate of the Past, 15, 1959–1983, 2019.*

*Lisiecki, L. E., & Raymo, M. E., A Pliocene-Pleistocene stack of 57 globally distributed benthic $\delta^{18}O$ records, Paleoceanography, 20, 1, 2005.*

Review of

Disparate energy sources for slow and fast Dansgaard-OEschger cycles

by Liebrand et al.

*We are thankful to the anonymous reviewer for reviewing our manuscript. We have found their critical review and suggestions to improve and clarify the manuscript very helpful.*

This paper presents a bispectral analysis of the NGRIP stable isotope record in order to identify possible energy transfers between two frequencies into their sum and differences. This allows the formulation of hypotheses regarding how two types of DO events may be triggered assuming periodic signals. The paper is well written and the conclusions, based on the mathematical/numerical analysis are sound. It is interesting because it uses a method that is not widely known in the climate dynamics community (well known, however in the fluid dynamics community). Where it comes to the physical interpretation of the results, this study -- by necessity -- remains speculative. Some more analysis, in particular of available model simulations, would make this paper a valuable contribution to Climate of the Past.

Major Comments:

1) This study is based on the premise that the climate system is principally a non-linear oscillator which is forced by very slow processes (essentially the Milankovic cyles) or very short cycle (solar cycles) and very fast processes (solar cycles), and from which energy is fed into the "DO band". This premise should be stated upfront and much more clearly.

*We agree that this is indeed our premise, and that we had not clearly indicated this in our introduction. As Reviewer 2 points out, we must indeed consider the possibility that the asymmetric DO signals in the NGRIP $\delta^{18}O$ record, which we deconvolve using bispectral analysis, are -not- resulting from nonlinear frequency interactions, but by coincidence contain a bispectral structure that supports the notion (our notion) that energy is transferred from the astronomical periodicities to those of slow-DO cycles, and from the centennial periodicities to those of fast-DO cycles. We do not believe the documented interactions (e.g., Zone 3, Fig. 2) to be a sheer coincidence, but agree with Reviewer 2 that the bispectrum alone cannot rule this option out. It, therefore, forms our premise, which we will state upfront in the revised version of this paper.*

It is certainly in contrast with the mainstream view that DO variability is not dominantly cyclical and that the major changes are likely forced by ice sheet-ocean instabilities and mediated by AMOC changes. While this premise can certainly be justified for this study, I suggest that the authors provide a more detailed discussion of alternative mechanisms that most likely provoked DO variability.

*Our results do not necessarily contrast with previously proposed mechanisms related to ice-sheet/ocean instabilities mediated by AMOC changes. In fact, we include several of these mechanisms into our discussion already (see Section 4). We believe that the addition of this*

*study is that the pacing and/or rate of inceptions and terminations of DO stadials and interstadials is nonlinearly paced (at least in part) by non-millennial forcing agents (i.e., astronomical and solar). In other words, the timings and structure of AMOC mediated ice-sheet/ocean instabilities is indirectly/nonlinearly determined, in part, by forcings that operate on shorter and longer time scales.*

*Reviewer 1 also commented on us strengthening our discussion of potential mechanisms. We will take this criticism on board and improve Section 4 of the manuscript.*

They should also critically reflect on the fact that the various solar cycles on the fast time scale end provide insufficient amplitude to force the slower components appreciably. Significant amplifying feedbacks would be required to trigger DO events from that end.

*We partially agree with Reviewer 2 on this point and already mention the carbon cycle as a potential positive feedback mechanism (lines 184-185). The carbon cycle could also function as an energy redistributing mechanism, for example by chemically storing energy, and releasing it at a later point in time. We will incorporate this point into the discussion.*

*One comment regarding "the various solar cycles on the fast time scale end provide insufficient amplitude to force the slower components appreciably": it is important to realise that the energy transferred within a triad (i.e., an interaction among three frequencies) is described by both amplitude and frequency. Furthermore, it is crucial to consider frequency during the integration over the bispectrum (Fig. 3 and 4), and to multiply with a coupling coefficient, to obtain energy conservation within each triad interaction. The coupling coefficient redistributes relatively more energy to the higher frequencies, as their recurrence is higher (more cycles completed) within the same time as a typical lower frequency. The success of achieving energy conservation, expressed per bispectral zone in this instance, is shown in Supp. Fig. S3.*

2) The authors filter the NGRIP data before bispectrally analysing it. The window width of the filter should be reported in the main text, so that the reader appreciates which variability is suppressed.

*Please see our reply to Point 4) in which we address this issue in full.*

3) Lines 170-180: I find this para rather speculative and the authors could make a much more convincing case if they presented the same analysis on modelled time series. Since their study is firmly based in the "oscillatory" world of DO interpretation (they do acknowledge this controversy), they could take a long time series from the simulations by Vettoretti et al (2022, Nature Geoscience) or Amstrong et al (2022, Climate Dynamics) which would be most suitable as these models exhibit self-sustained oscillations. It seems that the complex machinery that has been developed by the authors could be applied in a relatively straightforward manner to model data.

*We appreciate this comment and agree that further applications to model output would be valuable. In fact, such studies are currently being designed. However, new applications*

*and interpretations are beyond the scope of the current study because they would require more coding, as well as applying, interpreting, and presenting (in both text and figures) completely new bispectral results on model output, and the comparison of these new results to the NGRIP $\delta^{18}O$ results presented here. This would be the topic of a separate study.*

*We will strengthen the current discussion (lines 170 – 180) by including the cited literature above.*

4) Sensitivity of the conclusions to the selection of the filter that is used before spectrally analysing the data is missing. For example, how robust is the energy transfer from the short time scales with periodicities of <500 years? I suspect that energy transfer from these fast time scales is particularly sensitive to the specifics of the filter.

*We have not filtered the data before bispectral analysis, but (i) we have filtered the data prior to Wavelet analysis, (ii) we have used a Hamming tapering function prior to bispectral analysis, and (iii) we have used a coupling coefficient when integrating the bispectrum. All three processing steps were explained in the Appendix B, but we will briefly recap them here.*

*(i) Prior to the wavelet analysis (but not bispectral analysis), we applied a Gaussian notch filter to the NGRIP $\delta^{18}O$ record to suppress the longer, astronomical periodicities, and make sure we highlighted both astronomical and DO variability in equal measures in the wavelet and global spectrum (see lines 208–211, Fig. 1). We will add a sentence to main text of the revised paper that explains the effect of the Gaussian notch filtering on the wavelet results.*

*(ii) The Hamming tapering function/window, applied to the NGRIP $\delta^{18}O$ record prior to bispectral analysis, is unlikely to affect the higher frequencies. The higher frequency cycles pass nearly unaffected, especially in the central part of the record. Furthermore, there is not much DO variability in the Eemian and Holocene to be suppressed by the Hamming window. However, Hamming tapering does affect periodicities that are as long as the Hamming window itself (i.e., 123 kyr, in this instance). We note that the effects of Hamming tapering were already explained in Appendix B (lines 238–244), but we will add a clarification of this point to the main text and add the Hamming window length to both the main text and Appendix B.*

*(iii) The coupling coefficient, which indeed scales energy transfers with frequency, is part of the total and zonal integration step of the bispectral analysis (see 220–224, and 257–262, Fig. 3 and Fig. 4). In addition to explaining this in Appendix B, we will explain the effects of the coupling coefficient on the energy transfers we compute in the main text. We will also add the reasons for why a scaling with frequency is needed, when integrating over the bispectrum, to the main text. (See also our reply to Major Comment 1, part 3, above).*

*In summary, we will add explanations of the effects of all three data processing steps on the wavelet and bispectral results in the main text of the revised manuscript.*

5) In order to increase accessibility of the bispectral analysis to a wider readership, the article by Schmidt (doi: 10.1007/s11071-020-06037-z) could be cited.

*We will include this article in our discussion and add it to the reference list.*